# Hemiphosphoindigos as a platform for chiroptical or water soluble photoswitching

Fabien Kohl, Theresa Vogl, Frank Hampel & Henry Dube ✉

Photoswitches are important molecular tools to precisely control the behavior of matter by using light irradiation. They have found application in virtually all applied chemical fields from chemical biology to material sciences. However, great challenges remain in advanced property design including tailored chiroptical responses or water solubility. Here, hemiphosphoindigo (HPI) photoswitches are presented as capable phosphorus-based photoswitches and a distinct addition to the established indigoid chromophore family. Phosphinate is embedded in the core indigoid chromophore and the resulting optimized photoswitches display high thermal stabilities, excellent fatigue resistance and high isomer enrichment. A series of planar, twisted and heterocyclic HPIs are investigated to probe design strategies for advantageous photophysical properties. The phosphinate provides a platform for easily accessible, water-soluble photoswitches, especially interesting for biological applications. Its chiral nature further allows light-induced modulation of chiroptical properties. HPIs therefore open up a distinct structural space for photoswitch generation and advanced light-responsive applications.

Photoswitchable molecules enable reversible bottom-up light-control over nanoscopic behavior[1,2]. They have found their way into all chemistry-related fields ranging from molecular machines[3–5] or supramolecular chemistry[6–8] to biology[9–14], catalysis[15,16], or materials[17–20]. A number of different photoswitch motives are known but currently great efforts are being made to improve properties[21–23] or discover new structural realms and capacities of photoswitching. Such progress is urgently needed not only to improve performance but ultimately to open up diverse opportunities for switching and applications not possible yet. Examples encompass hydrazones[24–26], imidazolyl-radicals[27–29], diazocines[30–33], hetero-cyclopentane-1,3-diyls[34,35], homoaromatics[36], diphosphenes[37], Stenhouse dyes[38–42], or hemipiperazides[43] to name a few. Our group has concentrated its efforts on indigoid chromophores[44–48] and reported distinct applications, which were enabled by the specific chemical nature and photochemistry discovered for these compounds. Especially hemithioindigo (HTI)[44,49,50], hemiindigo (HI)[51–54], indirubin[55], peri-anthracenethioindigo (PAT)[56], rhodanine[57], or triox-obicyclononadiene (TOND)[58] provide outstanding possibilities for light-induced nanoscale control covering applications in molecular machine[59–67], supramolecular[65,68], material[54,56], or information processing[49,69] research. Other groups have joined the efforts broadening structure scope to e.g. aurones[70,71] and applicability, e.g. to enter the biological realm[47,72–80], further establishing indigoids as highly capable and broadly applicable photoswitches in general[81–94].

In this work, we present a class of phosphorus-based indigoid photoswitches, incorporating phosphinate as the central motif. The resulting hemiphosphoindigos (HPIs) are comprised of a phosphoindigo fragment connected to a stilbene fragment (see Fig. 1). The effects of different stilbene fragments on photoswitching behavior are analyzed in detail and parameters such as donor strength, planarity/twisting of the stilbene unit, heterocyclic substitution and hydrogen bonding are investigated. In this way, advantageous properties and guidelines for their rational design are established for HPI photoswitches. With regard to potential applications, the inherent stereogenic centre of the phosphinate was utilized for light-induced modulation of chiroptical responses. Late-stage hydrolysis of the phosphinate to the corresponding phosphinic acid introduces a

Friedrich-Alexander Universität Erlangen-Nurnberg, Department of Chemistry and Pharmacy, Nikolaus-Fiebiger-Str. 10, 91058 Erlangen, Germany.
✉ e-mail: henry.dube@fau.de

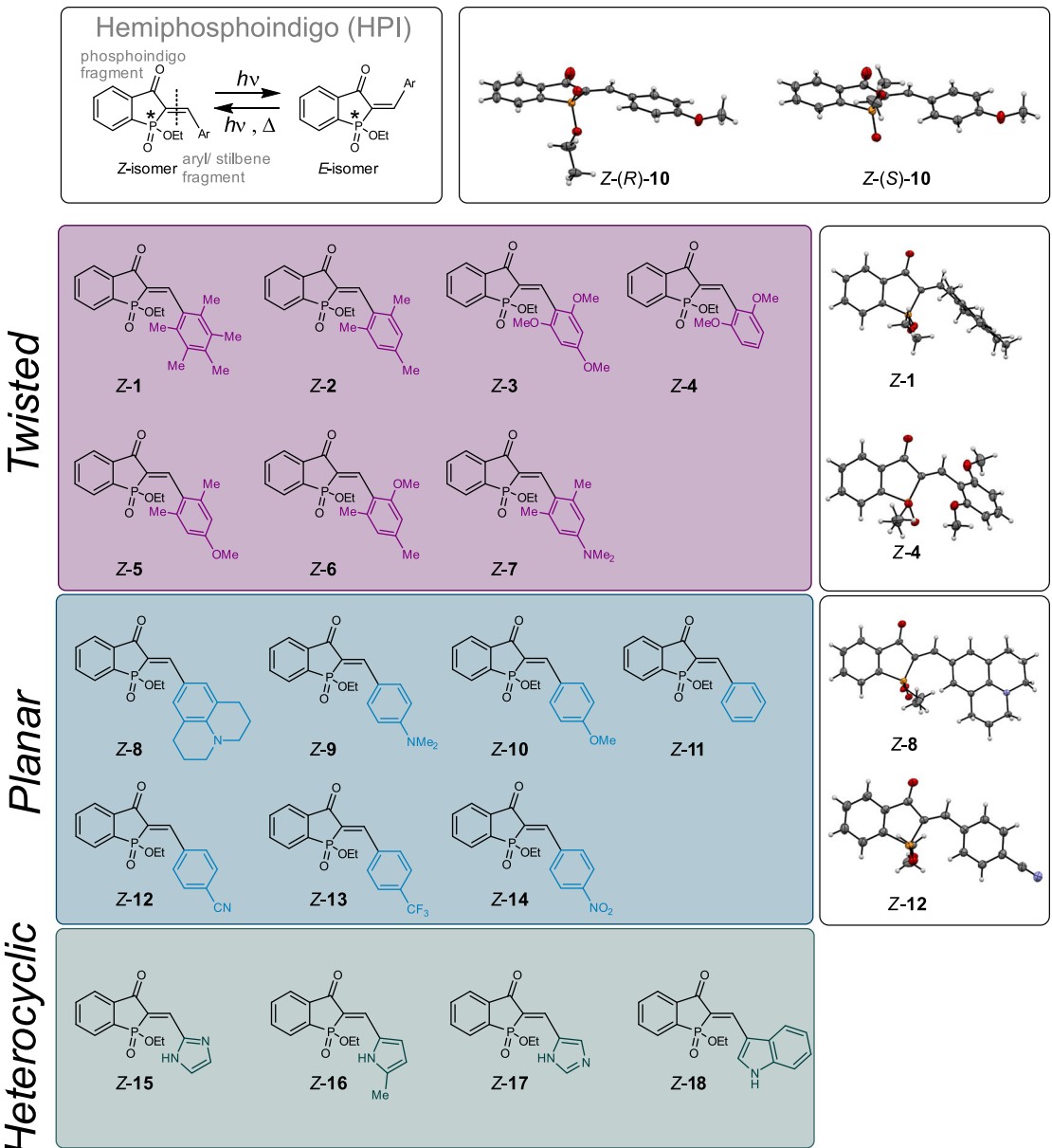

**Fig. 1 | HPIs 1-18 investigated in this study.** HPIs **1-7** adopt a twisted geometry because of steric hindrance induced by their *ortho*-substituents. HPIs **8-14** are fully planar and HPIs **15-18** contain a heterocycle with hydrogen bonding capacity. Twisted derivatives are shown on purple background, planar derivatives on blue and heterocyclic derivatives on dark green. Selected corresponding structures in the crystalline state of HPIs **1, 4, 8**, and **12** are depicted to illustrate the geometry differences, while the enantiomeric pure structures of planar *Z*-(*R/S*)–**10** highlight the chiral nature (for all structures determined in the crystalline state see Supplementary Note 16).

negative charge resulting in very good water-solubility while retaining proper switching capability.

## Results and discussion

A library of 18 HPI photoswitches was scrutinized, which can be classified in twisted- (derivatives **1-7**), planar- (derivatives **8-14**) and heterocyclic HPIs (derivatives **15-18**) as depicted in Fig. 1. The twisted HPIs possess different di-*ortho* substitution patterns and donors of varying strength in *para* position, while the planar HPIs bear a range of different electron-donating and withdrawing groups in *para* position. No acceptor substituents were introduced into twisted derivatives as acceptor groups generally proved to be detrimental for the performance of hemiindigoid photoswitches[50]. For the heterocyclic derivatives imidazole, pyrrole, and indole heterocycles with hydrogen-bonding capacity were employed since previous studies of *Arai*[95–97],

*Newhouse*[81], and our group[50] identified their highly advantageous effects in related HI and HTI photoswitches. To the best of our knowledge only HPI **11**[98] has been described previously. The HPIs are obtained after piperidine-catalyzed condensation of literature-known 1-ethoxy-2-hydrophosphindol-3-one 1-oxide[99] with the corresponding aryl aldehydes in yields of 28–96%, following established procedures for HTI photoswitches (for details see Supplementary Note 2, Supplementary Fig. 1). It should be mentioned that the synthesis of the related phosphoindigo[100,101] was previously reported but the photochemical behavior was not investigated. Crystals suitable for X-ray structural analysis were obtained for HPI derivatives *Z*-**1-6** and *Z*-**8-10, 12, 13** (see Fig. 1 and Supplementary Note 16). For HPI *Z*-**10**, enantiomeric pure crystals were obtained for each enantiomer after separation with chiral HPLC, which allowed direct assignment of the stereogenic center. The thus obtained molecular structures for HPIs **1,**

**2**, **4**, **5** and **6** clearly evidence the twisted spatial arrangement of up to 90° of the stilbene fragment with regard to the phosphoindigo fragment. *Z*-**3** adopts nearly planar geometry in the crystalline state but it can be assumed that is also twisted in solution similar to the related HTI structure[102]. In compounds *Z*-**8**-**14**, lacking the sterically encumbering di-*ortho* substitution, both stilbene and phosphoindigo moieties are coplanar. A coplanar arrangement of the heterocycles is also most probable because of their small steric hindrance (for geometry optimized structures see Supplementary Note 17 and Supplementary Data 1). Evidencing planar or twisted conformations in solution could be done by observation of the aromatic ring-current effect on the chemical shifts of the phosphinate ethyl-group in the corresponding $^1$H NMR spectra. For planar derivatives the CH$_2$-group signals are observed in the range of 4.05 ppm to 3.90 ppm while for twisted derivatives these signals are shifted significantly upfield to 3.86 ppm -3.43 ppm. The same behavior is seen for the CH$_3$-group signals (Supplementary Note 8, Supplementary Table 8). Furthermore, the ethyl-group signals of twisted HPI derivatives only are shifted upfield in the *Z* isomers, while for their corresponding *E* isomers similar shifts are observed as for planar HPI derivatives overall. This latter fact can easily be explained by the molecular geometry differences between *Z* and *E* isomers. Only in the *Z* configuration of twisted HPIs it is possible to position the phosphinate ethyl group close enough above the aromatic plane to experience the ring current.

The (photo)physical properties were analyzed for all derivatives and are summarized in Table 1. First, the thermal behavior was scrutinized. For all HPIs, the synthesis exclusively or dominantly affords the *Z* isomer as the thermally more stable isomer, which was evidenced by X-ray structural analysis or NOE experiments. Upon light irradiation, the metastable *E* isomer can be enriched and undergoes thermal isomerization to the *Z* isomer in the dark at elevated temperatures. After a certain time, a thermal equilibrium is reached between both isomers of HPIs **1**-**18**, unlike the near quantitative isomerization for the related HTIs. Kinetic analysis (Supplementary Note 4) afforded the corresponding *Gibbs* energies of activation for the *E* to *Z* isomerization $\Delta G^{\ddagger}_{E \rightarrow Z}$. The twisted HPIs exhibit exceptionally high thermal stability with $\Delta G^{\ddagger}_{E \rightarrow Z}$ ranging from 26.5 kcal mol$^{-1}$ to >35.5 kcal mol$^{-1}$. Due to higher order kinetics for the thermal isomerization of derivatives HPI **1** and **2** (see Supplementary Figs. 5 and 7), energies of activation were estimated, nevertheless the experimentally determined half-lives at 130 °C suggest even higher $\Delta G^{\ddagger}_{E \rightarrow Z}$ values in these cases. Unsurprisingly, the introduction of a strong donor group such as NMe$_2$ leads to stronger electron delocalization, hence destabilizing the central double-bond, which results in a lower barrier for thermal isomerization. This trend is also demonstrated by planar HPIs **8** and **9**, possessing significantly lower energies of activation compared to planar HPIs **10**-**14** with $\Delta G^{\ddagger}_{E \rightarrow Z}$ values greater than 27.6 kcal mol$^{-1}$. The $\Delta G^{\ddagger}_{E \rightarrow Z}$ values for the heterocyclic HPIs are slightly smaller with energies ranging between 22.2 kcal mol$^{-1}$ and 29.5 kcal mol$^{-1}$ compared to the twisted HPIs. Nevertheless, these derivatives still possess intermediate to very long thermal half-lives of their metastable *E* isomers.

After suitable thermal bistability was established, the photophysical and photochemical properties of HPIs **1**-**18** were analyzed (see Table 1 for all quantified values), the absorption and photoswitching behavior of one member of each of the three derivative classes is illustrated in Fig. 2. Key differences between the classes can be established for molar absorption coefficients, band separation of the *Z*- and *E*-isomers, and isomer enrichment upon light irradiation.

While both planar and heterocyclic HPIs display large molar absorption coefficients ranging from 15,500 to 62,500 L mol$^{-1}$ cm$^{-1}$ and from 21,700 to 46,300 L mol$^{-1}$ cm$^{-1}$, respectively, the ones for the twisted HPIs **1**, **2**, **5**, **6**, and **7** are significantly lower with values between 3800 to 16,000 L mol$^{-1}$ cm$^{-1}$. This behavior is explained by the twisted nature of the stilbene fragment with regard to the phosphoindigo core, which breaks conjugation between the two molecular parts. According

to this effect, derivatives **3** and **4**, bearing the sterically less demanding di-*ortho* methoxy-substitution, exhibit a lesser degree of twisting and therefore possess higher molar absorption coefficients. The molecular structures in the crystalline state of **3** and **4** confirm the less-twisted geometry. For the planar HPIs, which are *para*-substituted with different electron donating and withdrawing groups, the expected trend was confirmed. When increasing the donating strength of the substituent, the absorption band of lowest energy is gradually red-shifted, while the molar absorption is simultaneously increasing. This is also the case for the heterocyclic HPIs **15**-**18**.

Compared to related photoswitches of the indigoid family, namely HTIs[50,81,102–105] and HIs[51,53,74,95,97], the absorption of HPIs are significantly blue-shifted (see Supplementary Note 5, Supplementary Figs. 47 and 48). This can straight forwardly be explained by the higher oxidation state of the phosphorus, which acts like a second acceptor next to the carbonyl group of the phosphoindigo fragment. A similar hypsochromic effect is seen when HTIs are oxidized at the sulfur to the corresponding sulfoxide or sulfone[106]. With comparable size of the conjugated aromatic system and substituents, the difference in push-pull effects within the chromophores explains these findings. If HTI is not oxidized, the sulfur can effectively use its lone pairs as additional electron donors to the existing donor (stilbene fragment)–acceptor (carbonyl) system, consequently bathochromically shifting the whole absorption. In the present case, the phosphorus in the phosphinate motif is pentavalent and can only act as second acceptor.

Striking differences in absorption band separation of the corresponding *Z* and *E*-isomers are observed for the three different classes and showcased in Fig. 2a for selected examples. Photoswitching experiments were monitored via UV/vis spectroscopy and $^1$H NMR spectroscopy and carried out in toluene solution using light of different wavelengths until a stable isomer concentration was reached in the photostationary state (pss). The planar HPIs **8**-**14** display virtually no distinguishable absorptions bands for the two different isomers, which leads to low *E* isomer enrichment ranging from 31% to 65% upon light irradiation. Contrary, for the twisted HPI derivatives up to 88% of *E* isomer can be enriched and the reverse *E* to *Z* photoisomerization proceeds smoothly under irradiation with light of longer wavelengths recovering more than 90% of the *Z* isomer. Thus, substantially improved switching capacity is achieved for the twisted HPI derivatives compared to their planar counterparts. This very good performance is possible because of a bathochromic tailing of the *E* isomers absorption in conjunction with much higher quantum yields for the *Z* to *E* photoisomerization (37-52%) as opposed to the *E* to *Z* photoisomerization (1-24%). The quantum yields are quite high overall and thus allow for very fast and efficient photoswitching within seconds to minutes under typical low- and medium power LED irradiation at UV/vis concentrations (see Supplementary Note 8, Supplementary Figs. 69–86). Generally lower quantum yields for *E* to *Z* photoisomerization as opposed to *Z* to *E* photoisomerization are also found in HTIs and hint at a similar excited state landscape for the *E* isomers in HTIs and HPIs[100]. The only exception is HPI **7**, which possesses low single digit quantum yields for both photoisomerizations. In this case the combination of a strong electron-donation and pre-twisting of the stilbene fragment in the ground state could well facilitate formation of a twisted intramolecular charge transfer (TICT) state. Such TICT state would offer a competing deexcitation channel diminishing the quantum yields for photoisomerization, similar to what is well evidenced for structurally related HTI photoswitches[99].

Furthermore, twisted HPI **3** could be alternated for 50 irradiation cycles with 395 and 470 nm, showing high resistance to photofatigue (Fig. 3a). In general, all HPI derivatives with exception of the planar derivative **10** and the ones bearing electron withdrawing groups **12**-**14** display virtually no signs of photodegradation after 10 irradiation cycles (Supplementary Note 12, Supplementary Figs. 115–123). In case of the heterocyclic HPIs **15**-**17**, significant band separation can be

**Table 1 | Overview of the photophysical and physical properties of HPIs 1-18. All experiments were performed in toluene-$d_8$ or $p$-xylene-$d_{10}$ unless stated otherwise**

| HPI | $\Delta G^{\ddagger}_{E \to Z}$ [kcal mol$^{-1}$] | Half-life of $E$ isomer at 25 °C[a] | $\Delta G$ [kcal mol$^{-1}$] | $\phi_{Z \to E}$ & $\phi_{E \to Z}$ | Isomer % in pss (at irr. wavelength) | $\varepsilon_{max}$ [L mol$^{-1}$ cm$^{-1}$] and λ [nm] at $\varepsilon_{max}$ of Z and E isomers |
|---|---|---|---|---|---|---|
| **1** | >35.5 | >380000 a | >1.5[b] | 50% & 1% | 88% $E$ (395 nm) 98% $Z$ (450 nm) | $Z$: 341; 3800 $E$: 344; 5500 |
| **2** | >35.0 | >165000 a | >1.8[b] | 40% & 10% | 80% $E$ (300 nm) 100% $Z$ (450 nm) | $Z$: 330; 5300 $E$: 338; 8500 |
| **3** | 28.8 | 4 a | 2.4 | 52% & 24% | 82% $E$ (395 nm) 98% $Z$ (470 nm) | $Z$: 391; 29,600 $E$: 377; 11,800 |
| **3-OH** | 21.1 | 6 min | 1.8 | 9% & 7% | 63% $E$ (405 nm) 92% $Z$ (470 nm) | $Z$: 403; 23,300 $E$: 414; 11,000 |
| **4** | 30.3 | 50 a | 1.9 | 47% & 21% | 79% $E$ (395 nm) 99% $Z$ (450 nm) | $Z$: 358; 22,400 $E$: 346; 12,500 |
| **5** | 33.5 | 11500 a | 1.7 | 37% & 9% | 68% $E$ (395 nm) 98% $Z$ (450 nm) | $Z$: 357; 5500 $E$: 358; 8700 |
| **6** | 30.0 | 30 a | 1.6 | 45% & 23% | 81% $E$ (395 nm) 98% $Z$ (470 nm) | $Z$: 339; 10,800 $E$: 334; 8100 |
| **7** | 26.5 | 33 a | 1.4 | 3% & 2% | 47% $E$ (430 nm) 91% $Z$ (530 nm) | $Z$: 449; 9100 $E$: 481; 16,000 |
| **8** | 22.7 | 1.4 h | 2.3 | 21% & 22% | 31% $E$ (470 nm) 99% $Z$ (565 nm) | $Z$: 485; 62,500 $E$: 495; 53,100 |
| **9** | 22.0 | 25 min | 3.1 | 26% & 23% | 41% $E$ (450 nm) 93% $Z$ (505 nm) | $Z$: 456; 53,600 $E$: 464; 46,900 |
| **10** | 29.0 | 7 a | 3.1 | 51% & 18% | 65% $E$ (365 nm) 98% $Z$ (470 nm) | $Z$: 372; 25,800 $E$: 373; 26,400 |
| **11** | 27.6 | 210 d | 3.3 | 46% & 30% | 66% $E$ (340 nm) 99% $Z$ (450 nm) | $Z$: 332; 30,500 $E$: 334; 22,300 |
| **12** | 29.5 | 15 a | 2.6 | 48% & 30% | 60% $E$ (340 nm) 97% $Z$ (450 nm) | $Z$: 328; 27,600 $E$: 328; 24,900 |
| **13** | 27.7 | 260 d | 3.4 | 51% & 49% | 53% $E$ (340 nm) 98% $Z$ (450 nm) | $Z$: 322; 21,000 $E$: 328; 18,100 |
| **14** | 28.1 | 1.5 a | 2.9 | 36% & 59% | 53% $E$ (365 nm) 98% $Z$ (450 nm) | $Z$: 332; 18,000 $E$: 327; 15,500 |
| **15** | 24.0 | 3 d | 1.2 | 29% & 33% | 69% $E$ (365 nm) 99% $Z$ (450 nm) | $Z$: 383; 24,400 $E$: 414; 21,700 |
| **16** | 26.4 | 5 d | 0.7 | 12% & 14% | 66% $E$ (395 nm) 99% $Z$ (470 nm) | $Z$: 430; 46,300 $E$: 450; 38,100 |
| **17** | 29.5 | 14 a | 1.0 | 30% & 12% | 84% $E$ (340 nm) 100% $Z$ (450 nm) | $Z$: 372; 30,100 $E$: 402; 22,000 |
| **18** | 22.2 | 34 min | 2.6 | 35% & 11% | 63% $E$ (430 nm) 60% $Z$ (470 nm) | $Z$: 424; 23,700 $E$: 430; 23,400 |

[a]Linearly extrapolated. [b]Value estimated before full completion of thermal isomerization, since the experiment is very slow and not finished within a reasonable time.
For hydrolyzed HPI **3-OH**, all experiments were performed in D$_2$O.

achieved, which is in accordance with the previously reported heterocyclic HTIs[50]. Nevertheless, very high $E$ isomer enrichment for heterocyclic HPIs is only possible for derivative **17**, possibly due to the fact that the phosphinate adds a second hydrogen bond acceptor rendering the $Z$ isomers stabilized in this class. The importance of fine-tuned intramolecular hydrogen bonding for advantageous switching behavior in HPIs is further highlighted by 3-indol derivative **18**. In this case, intramolecular hydrogen bonding is not possible because of the unfavorable geometry leading to poor band separation and consequentially low isomer enrichments in both directions. Overall, very good photoswitching properties are obtained for twisted HPIs and for imidazole-derivative **17**, which delineates important design criteria for future elaboration of this class of photochromes.

Due to the inherent chiral nature of the phosphinate group within HPIs, their chiroptical properties were also investigated. The ($R$) and ($S$) enantiomers of planar HPI $Z$-**10** and the twisted HPIs $Z$-**3** and $Z$-**6** were separated by chiral HPLC and the corresponding ECD spectra were measured in toluene solution (Fig. 3b–d). ECD signal intensity for both planar and twisted derivatives is moderate due to the electronic similarity of the oxygen and the ethoxy group at the phosphorus

chirality center. A slight increase of the signals can be observed for twisted HPIs. Introduction of another stereo-information using a non-symmetric stilbene fragment in the case of HPI **6** leads to appreciable increase in signal intensity (Fig. 3). Irradiation of enantiomerically pure $Z$-**6** with 395 nm induces $Z$ to $E$ photoisomerization, resulting in pronounced ECD changes and a reversal of the Cotton-effect throughout the spectrum (Fig. 3d). This process is reversible and the initial state can be recovered upon irradiation with 470 nm, which establishes an advantageous stable, chiroptical photoswitching with large modulation of the signal.

An accompanying theoretical description on different DFT levels of theory was delivered for HPIs **3, 6**, and **10** covering planar and twisted geometries (Supplementary Note 7). From the calculations it becomes evident that planar structures like HPI **10** are well modeled by theory allowing to well reproduce absorption and ECD spectra. The latter allows for facile determination of absolute stereo-configuration for the different isomers. However, twisted HPIs are more problematic in the theoretical description as the twist of the stilbene fragment has a significant influence on the ECD signal. Since differently twisted structures are energetically close, their individual contributions to the

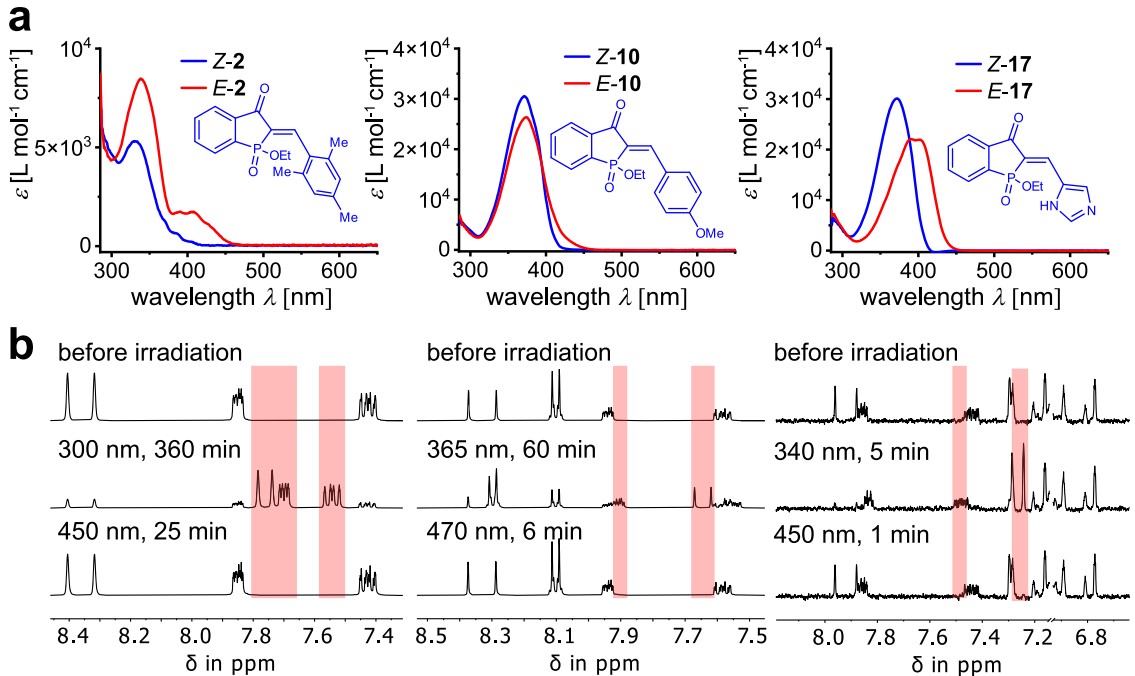

**Fig. 2 | Absorption and photoswitching behavior of twisted HPI 2, planar HPI 10, and heterocyclic HPI 17. a** Molar absorption coefficients of HPIs **2, 10**, and **17** in toluene solution for both pure isomeric forms. **b** Photoswitching of HPI **2, 10**, and **17** followed by $^1$H NMR spectroscopy in toluene-$d_8$. Spectra were recorded after light irradiations when the respective pss was reached. The parts in the spectra highlighted in red show signal regions of the corresponding *E* isomers. The spectra termed "before irradiation" were obtained from pure separated *Z* isomers.

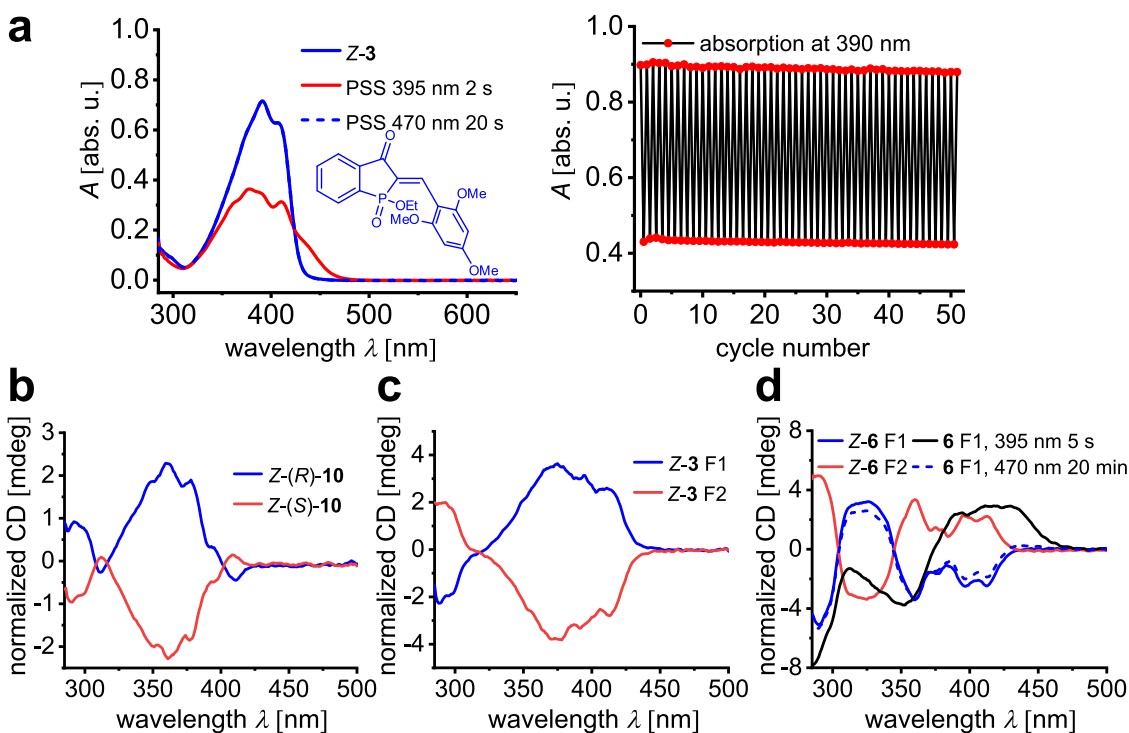

**Fig. 3 | Photofatigue resistance of HPI 3 and ECD spectra of HPIs 10, 3 and 6. a** (left) Absorption spectrum of twisted HPI *Z*-**3** upon consecutive light irradiation with 395 nm and 470 nm. (right) Change of absorption of HPI **3** at 390 nm during 395/470 nm irradiation cycles. **b** Experimental ECD spectra for the two enantiomers of planar HPI *Z*−**10**. **c** Experimental ECD spectra for the two enantiomers of twisted HPI *Z*-**3**. **d** Experimental ECD spectra for the two enantiomers of twisted HPI *Z*-**6** and after irradiation with 395 nm and 470 nm light. All spectra are measured in toluene and enantiomers are assigned by comparison with crystal structures of enantiomeric pure HPIs, or numbered corresponding to their elution order on chiral HPLC.

**Table 2 | Overview of photoswitching properties of HPI 1-18 and hydrolyzed derivatives 1-OH, 2-OH, 3-OH, 11-OH and 17-OH in water and D$_2$O**

| HPI | T [°C] | $\lambda_{irr}$ E to Z [nm] | [E] at PSS [%] | $\lambda_{irr}$ Z to E [nm] | [Z] at PSS [%] | Photochromism (no=X or good=✓), additional comments |
|---|---|---|---|---|---|---|
| 1 | - | - | - | - | - | X, not soluble |
| 1-OH | 23 | 365 | 70 | 430 | 100 | ✓ |
| 2 | 23 | 365 | - | 430 | - | ✓, not soluble enough for NMR |
| 2-OH | 23 | 365 | 63 | 430 | 100 | ✓ |
| 3 | 23 | 385 | - | 470 | - | ✓, not soluble enough for NMR |
| 3-OH | 6 | 405 | 63 | 505 | 92 | ✓ |
| 4 | 23 | 365 | n.d. | 450 | n.d. | ✓ |
| 5 | 23 | 365 | n.d. | 470 | n.d. | ✓ |
| 6 | 23 | 365 | n.d. | 470 | n.d. | ✓ |
| 7 | 23 | - | - | - | - | X |
| 8 | 23 | - | - | - | - | X |
| 9 | 23 | 530 | - | - | - | thermal isomerization too fast |
| 10 | 23 | 365 | n.d. | 470 | n.d. | ✓ |
| 11 | 23 | 365 | - | 450 | - | ✓, degradation |
| 11-OH | 23 | 365 | 78 | 430 | 100 | ✓ |
| 12 | 23 | - | - | - | - | X |
| 13 | 23 | - | - | - | - | photodegradation, thermal degradation |
| 14 | 23 | - | - | - | - | irreversible photoreaction |
| 15 | 23 | 365 | 18 | 450 | 89 | ✓ |
| 16 | 23 | - | - | - | - | X |
| 17 | 23 | 340 | 42 | 450 | 99 | ✓ |
| 17-OH | 23 | 385 | 52 | 430 | 84 | ✓ |
| 18 | 23 | - | - | - | - | X |

sum-spectra cannot be predicted with certainty because of the larger energy-error margin of DFT calculations (typically within 1-2 kcal mol$^{-1}$).

Since HPIs all bear a polar phosphinate group we tested all of them for their inherent water solubility. A full summary of the measured solubilities can be found in Table 2 and the Supplementary Information (Supplementary Note 3). We found substantial water-solubility for most HPIs but especially for **15** (0.25 g L$^{-1}$) and **17** (0.15 g L$^{-1}$). For this reason, the photoswitching of the latter two HPIs could be quantified by NMR spectroscopy. All other HPIs could only be investigated with UV/vis spectroscopy to establish photochromism and photoswitchability qualitatively. While **15** cannot be photoswitched to appreciable extent in water, HPI **17** shows viable photoswitching allowing to accumulate 42% of the metastable E isomer with 340 nm light and 99% of the Z isomer with 450 nm light.

We then projected that water-solubility of HPIs could be substantially improved by late-stage hydrolysis of the phosphinate group to the corresponding phosphinic acid using TMSBr (Fig. 4). Phosphinic acid derivatives themselves are of high importance in biology and for the treatment of diseases due to their biological activity, enabling, among others, their use as peptide and enzyme inhibitors[107,108]. Hydrolysis of HPIs could thus lead to additional benefits by generating potentially biologically active compounds in addition to increase in water solubility. We therefore chose the five HPI derivatives **1, 2, 3, 11**, and **17** for hydrolysis and in-depth characterization to cover all, twisted, planar, and heterocyclic, variants. At the same time the chosen derivatives showed the best photoswitching performances and thermal bistabilities in their respective class in organic solvents (or water

for their non-hydrolyzed form (Supplementary Note 9)) and were thus deemed the most promising candidates. The obtained results for hydrolyzed HPIs-OH are highlighted in Fig. 4 and summarized in Table 2 and Supplementary Note 10.

All hydrolyzed HPIs possess improved water-solubility compared to their non-hydrolyzed counterparts (see Supplementary Table 1, Supplementary Note 3), allowing analysis in pure D$_2$O via $^1$H NMR spectroscopy. Hydrolysis to the phosphinic acid only leads to small absorption spectral changes and the photoswitching capacity is typically well retained. The metastable E isomer can be enriched substantially in pure water reaching up to 78% for HPI **11-OH** or 70% and 63% for HPIs **1-OH** and **3-OH**, respectively. Likewise, very high enrichment of the corresponding Z isomers beyond 84% and oftentimes quantitatively is possible when irradiating at longer wavelengths. The quantum yields are diminished compared to the non-hydrolyzed HPIs but still sizable enough ($\phi_{Z\to E}$ = 9% and $\phi_{E\to Z}$ = 7% for **3-OH**) for fast photoswitching under typical LED irradiation conditions. The thermal E to Z isomerization was quantified by kinetic analyses for all hydrolyzed HPIs-OHs revealing lower Gibbs energies of activation for this process compared to the corresponding non-hydrolyzed HPIs (see Supplementary Table 2, Supplementary Note 4). While the Gibbs energies of activation for HPIs **3-OH, 11-OH** and **17-OH** now range between 21.1–26.7 kcal mol$^{-1}$, which corresponds to half-lives between minutes and days at 25 °C, measurements of HPIs **1-OH** and **2-OH** reveal very high thermal stability of their metastable states with estimated half-lives exceeding hundreds of years at ambient temperature. With this performance especially HPIs **1-OH, 2-OH** or **11-OH** are very well performing photoswitches in water. Although thermal bistability is significantly diminished in aqueous solution for HPI **3-OH**, its established switching properties are still well suited for many biological applications, especially ones requiring a self-deactivation of the photoswitch (for a discussion of thermally labile photoswitches for biological applications see for example refs. 109–111 and literature cited therein). Likewise, for dissipative applications in water, where a short-lived high-energy metastable state is required to thermally dissipate its energy at fast timescales, the behavior of HPI **3-OH** would also be ideally suited.

Overall, many HPIs showed already enough water solubility to allow qualitative demonstration of their photoswitching capacity in this medium in their non-hydrolyzed form. After hydrolysis to the corresponding phosphinic acids water solubility is improved significantly and very good photoswitching properties are obtained for derivatives **1-OH** and **11-OH** in particular. Both derivatives allow high isomer enrichment in the respective pss and show high thermal stability for full light-control of their isomerizations and thus are well competitive with the state of the art in water-soluble photoswitching[112].

In summary, we present a class of phosphorus-based indigoid photoswitches where the best performing derivatives deliver high thermal stabilities, excellent fatigue resistance, and very good switching performance. Key strategies for their enhanced photophysical properties include pre-twisting of the stilbene fragment or the introduction of imidazole-derived heterocycles with H-bonding capacity. Despite twisting leading to a disruption of the conjugated π-system, photochromism, addressability, and isomer enrichment are in fact enhanced. Contrary to other indigoid photoswitches, the introduction of electron-rich substituents or electron-rich heterocycles alone does not suffice to achieve advantageous photophysical properties. The phosphinate structure with its intrinsic stereogenic center allows light-induced modulation of chiral information, which could be used in the field of data storage[113–115] or sensing[116]. Further, the molecular framework allows facile late-stage conversion to water-soluble photoswitches with a biologically highly relevant functional phosphinic acid group, while maintaining very good switching properties for the best performers. Therefore, HPIs show high potential for the

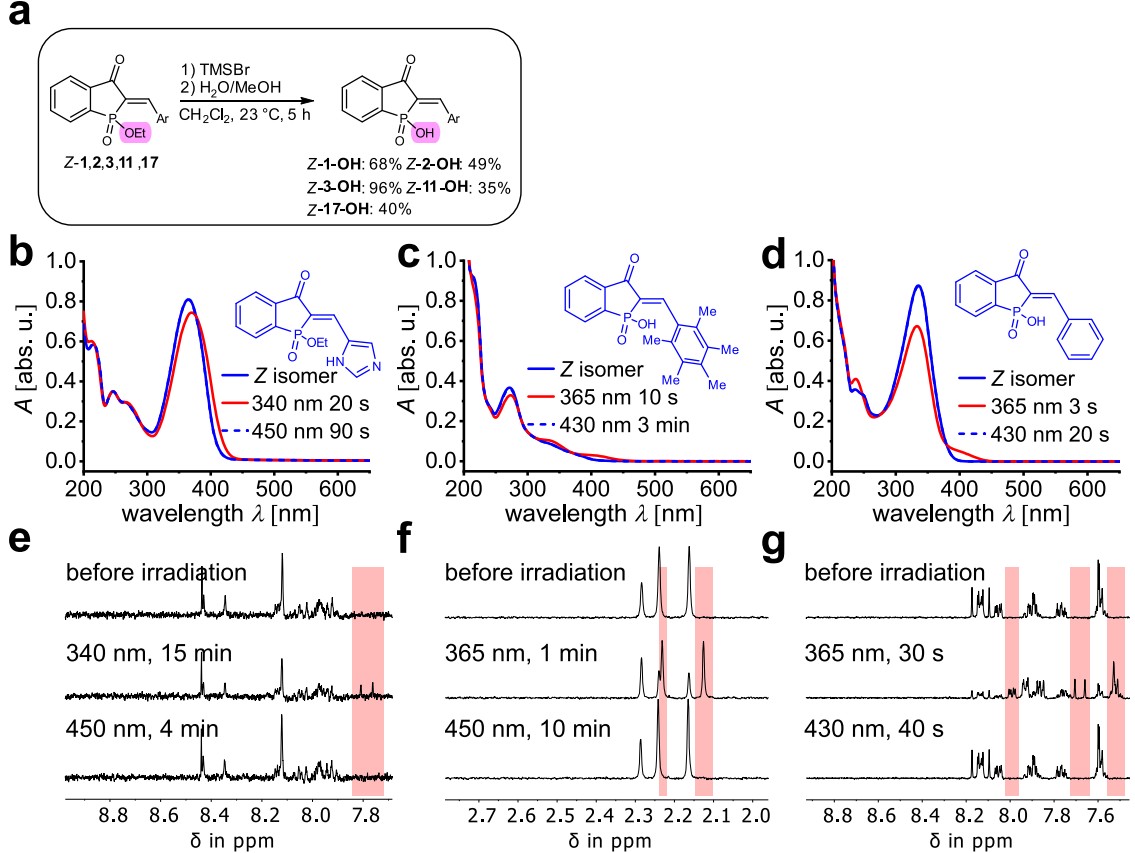

**Fig. 4 | Absorption and photoswitching behavior of HPI 17 and hydrolyzed HPIs in water or D$_2$O. a** General synthetic procedure for the hydrolysis of HPI **1, 2, 3, 11, 17** to the corresponding phosphinic acids. Substitution at the phosphorus are highlighted in purple. **b, c, d** Photoswitching of non-hydrolyzed HPI **17** and

hydrolyzed HPI **1-OH** and **11-OH** in water followed by UV/vis spectroscopy in water at 23 °C. **e, f, g** Photoswitching of non-hydrolyzed HPI **17** and hydrolyzed HPI **1-OH** and **11-OH** in water followed by NMR spectroscopy in D$_2$O at 25 °C. The parts in the spectra highlighted in red show signal regions of the *E* isomer.

development of light-activated drugs and other biological applications in the future.

## Methods
General materials and methods are described in Supplementary Note 1.

### Synthesis
Precursors ethyl 2-(ethoxy(methyl)phosphoryl)benzoate **19**[117], and 1-ethoxy-2-hydrophosphindol-3-one-1oxide **20**[99] were prepared according to adapted literature known procedures. Hydrophosphindol **20** was then condensed with different aldehydes in toluene using piperidine as catalyst to yield the HPIs presented in this work. Most aldehydes were commercially available, the synthesis for aldehydes **21-24** is outlined in Supplementary Fig. 3 and followed published protocols[102,118–120]. Hydrolysis to the HPI-OHs was performed in anhydrous CH$_2$Cl$_2$ using trimethylbromosilane. For more information on the synthesis and spectroscopic data see Supplementary Note 2, the NMR spectra are displayed in Supplementary Note 15.

### Thermal isomerizations
The thermal isomerizations of the metastable *E* isomers of the HPIs and HPI-OHs were investigated by charging NMR tubes with the respective *Z* isomers, as obtained from the synthesis, and adding toluene-*d*$_8$ or *p*-xylene-*d*$_{10}$ or D$_2$O. An *E* isomer enriched solution was obtained by irradiation with light of different wavelengths. The NMR tubes were then heated to the corresponding temperatures for isomerization reactions to occur and the kinetics were followed by $^1$H NMR measurements in defined time intervals. The conversion and isomer

concentration for each interval was determined by integration of well separated indicative proton signals. The kinetic data can be described by unimolecular first order reactions reaching equilibrium, the procedure is described in Supplementary Note 4 and the results are summarized in Supplementary Table 2.

### Determination of molar absorption coefficients of pure *Z* and *E* isomers
The molar absorption coefficients of pure *Z* and *E* isomers were determined from mixtures of both isomers. For this, a sample with known concentration was prepared and the isomer composition was determined by $^1$H NMR spectroscopy before recording a UV/Vis spectrum. After irradiation with light of a suitable wavelength, this step was repeated. The system of linear equations was solved using the NMR-spectroscopically quantified isomer compositions and the molar absorption coefficients for the pure isomer were calculated. For more details, see Supplementary Note 5.

### Quantum yield measurements
The quantum yields for both photoisomerization reactions (*Z* to *E* and *E* to *Z*) were determined using the instrumental setup of *Riedle* and coworkers[121]. Samples were prepared with absorptions ranging from 0.7 to 1.4. The samples were irradiated with light of a suitable wavelength close to the isosbestic point. After each irradiation step, the power of the solar cell detector and a UV/Vis absorption spectrum were recorded. Irradiation was continued until the photostationary state was reached. The setup allows for simultaneous determination of the quantum yields for both photoisomerization reactions by

calculating the concentration changes from the previously determined molar absorption coefficients and the recorded absorption spectra and performing a kinetic fit. For more information, see Supplementary Note 13, the results are summarized in Supplementary Table 10.

## ECD measurements

The concentration of samples for ECD measurements was adjusted until the absorption of the highest absorption band was between 0.8 and 1.0 (absorbance for optimal signal to noise ratio: 0.89). For better comparability, the concentration independent g factors were calculated: g factor= $\theta$/ Abs. For more details, see Supplementary Note 6.

## Theoretical description

DFT calculations were carried out using the Gaussian program package[122], and conformational searches were performed using the MacroModel[123] software package from the Schroedinger suite. Ground state geometries were optimized using different functionals and time-dependent DFT calculations were performed to obtain the theoretical UV/Vis absorption spectra and ECD spectra. Extraction of the spectral data and Boltzmann weighting was done using the SpecDis[124–126] program. More detailed information is given in Supplementary Note 7.

## Data availability

The data generated in this study are provided in the Supplementary Information file. Additional data generated during this study are available from the corresponding author H. D. upon request, from the Supplementary Data 1, and from the provided source data file. The X-ray crystallographic coordinates for the structures of HPI **1**-**6**, **8**, **9**, (*R*)-**10**, (*S*)-**10**, **12**, and **13** reported in this study have been deposited at the Cambridge Crystallographic Data Centre (CCDC), under CCDC numbers 2309092 (**1**), 2309091 (**2**), 2309093 (**3**), 2309094 (**4**), 2309098 (**5**), 2309100 (**6**), 2309099 (**8**), 2309097 (**9**), 2309101 ((*R*)-**10**), 2309102 ((*S*)-**10**), 2309095 (**12**), 2309096 and (**13**). These data can be obtained free of charge from the Cambridge Crystallographic Data Centre via www.ccdc.cam.ac.uk/data_request/cif. Source data are provided with this paper.

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

## Acknowledgements

H. Dube thanks the Deutsche Forschungsgemeinschaft (DFG) for an Emmy Noether fellowship (DU 1414/1-2, Gepris number 264598191). This project has also received funding from the European Research Council (ERC) under the European Union's Horizon 2020 research and innovation programme (PHOTOMECH, grant agreement No 101001794).

## Author contributions

H.D. and F.K. designed the project. F.K. synthesized all compounds with exception of HPI **3** and **7**, which were synthesized and characterized by T.V. who also performed preliminary (photo)physical analysis for these derivatives. F.K. conducted all thermal, photochemical experiments, analyzed the data and performed the theoretical calculations. F.H. determined the structures in the crystalline state. F.K. and H.D. wrote the manuscript.

## Funding

## Competing interests

The authors declare no competing interests.
