## [Transparent Peer Review file · Nature Communications]

Hemiphosphoindigos as a Platform for Chiroptical or Water Soluble Photoswitching

Corresponding Author: Professor Henry Dube

Version 0:

Reviewer comments:

Reviewer #1

(Remarks to the Author)

The manuscript by Dube and co-workers describes synthesis and photoswitching of hemiphosphoindigo derivatives. Importantly, this class of compounds has been described previously but the photoswitching has not been studied yet. In general, the manuscript could be interesting for the readership of Nat. Commun. and fitting to the scope of the journal. However, in my opinion, publication of this work is premature and further experiments are needed to make the manuscript ready for publication. Below are the points that I would like the authors to consider:

- 1) The statement "water soluble photoswitches" in the title and in the text is misleading. Only three out of 18 compounds have been checked for the photoswitching in water and two out of these three ones showed either bad (42% conversion in pss) or very bad (18% conversion in pss) performance in water. Additionally, the choice of compounds for the studies in water looks rather random (one twisted HPI, two heterocyclic ones, no planar ones...). To explore the photoswitching of HPI in water, the solubility and photochromism should be checked for all 18 compounds and honestly compared with the literature data on other classes of photoswitchable molecules in water (for collection of examples, see <https://doi.org/10.1039/D0CS00547A>).
- 2) Abstract: "... high isomer enrichment..." – this statement cannot be applied to the most of the studies compounds (<80% of E-isomer in PSS).
- 3) Lines: 44-46 and 142-143: the information on the recent studies of the photoswitching of aurones belonging to the indigoid family should be added and the corresponding publications should be cited.
- 4) Lines 63-65: could the authors explain why no acceptor groups were introduced in the twisted HPI?
- 5) It is not clear, whether all presented HPI are novel compounds or some of them have been reported previously. Please add this information to the text.
- 6) Since the photoswitching of HPI is reported for the first time, it is necessary to isolate and fully characterize the photoinduced Z-isomers. Considering the long lifetimes, this should be possible for most of the compounds. If this is done, the molar absorption coefficients and absorption spectra of the photoinduced isomers can be determined directly (cf. the calculations described in Section 1.4 of SI).
- 7) Line 175: "... very fast... photoswitching..." - please add the corresponding reaction times to Table 1.
- 8) Lines 256-259: "... fully suited for many biological applications..." Please provide examples of the relevant biological applications along with references.
- 9) Taking into account the above critical points, the conclusions are not fully supported by the results and should be revised.

Reviewer #2

(Remarks to the Author)

In this paper, the authors have explored a class of phosphorus-based indigoid photoswitches with high thermal stabilities, excellent fatigue resistance, and very good switching performance, which could be used in the field of data storage, sensing. Further, the molecular framework allows facile late-stage conversion to water-soluble photoswitches with a biologically group, while maintaining the switching properties. This has high potential for the development of light-activated drugs and other biological applications in the future. In my opinion, this work is sufficiently novel and suitable to be accepted by Nature Communication, but there are some problems in the manuscript that need the authors make changes after considering the following suggestions.

Suggestions noted below:

1. How to prove that Z-3 adopts twisted state in solution?
2. What is the relationship between Figures S3, S4 and the order of thermal isomerization as described in line 104?
3. What are the possible reasons for the high water solubility of Z-15 and Z-17?
4. Please provide the light irradiation time required for photoisomerization of the three photoswitches shown in Figure 2.
5. In the captions of Figures S3 and S4, it states, "the value is given in Table S1," but Table S1 contains data on water solubility. Please check them.
6. The geometry of HPis 15-18 should also be provided in Supporting Information for the reader's reference for comparison.
7. On line 73, why crystals sample suitable for X-ray structural analysis can't be obtained from HPI derivatives Z-6, 7, 11, 13-18? Whether it is the poor chemical stability of these products under X-rays or some other problem? Please the authors explain this in the main text.
8. On line 108, HPI 7 is not planar geometry. In addition, the comparison between HPI 7,8 and HPI 10-14 can't explain the trend authors referred before. Please the authors correct it.
9. On line 142-145, the data here are insufficient, and the authors should convert some of the examples from HTIs, HIs, and HPis into figure to more visually show the significant blueshift of HPis.
10. On line 245-267, the necessary figures to show the data are missing here. The data for HPI 15-OH and HPI 17-OH should be supplemented and the data for HPI 3,15,17, HPI 3-OH,15-OH,17-OH should be compared.
11. If possible, the authors provide a potential application case which could be used in the field of data storage or sensing.

Reviewer #3

(Remarks to the Author)

The manuscript describes an extensive series of new photoswitches comprising hemiphosphoindigo cores with varying substituents on the stilbene moiety. This further diversifies the extensive work this group has done on indigoid-type photoswitches. The short synthetic route enables a variety of derivatives to be made relatively easily. The synthesis and characterisation of the switches is presented clearly and thoroughly, and detailed photophysical and photochemical characterisation has been carried out to develop structure/property relationships for the various observations. Efforts have been made to support these conclusions with computational work, although the authors note that for some of the switches, conformational variation makes spectral prediction by TDDFT challenging.

The photoswitches are interesting new additions to the literature, and the added aspect of the chiroptical properties makes them an exciting area for further development.

I think the manuscript is suitable for publication, subject to a few minor clarifications.

Minor comments:

(1) Table 1 might benefit from extra columns to separate some of the data. The current layout is quite difficult to follow.

(2) Figure 2 and line 165 onwards – the authors say that planar HPis 8-14 have very little band separation hence poor Z to E conversion on irradiation. But for some compounds, this still seems to result in almost quantitative recovery of the Z form despite similar quantum yields for forward and reverse processes. e.g. HPI-8, the PSS achieved under 565 nm irradiation is almost quantitative in Z, despite the quantum yields for Z to E and E to Z being equal. There appears to be no appreciable absorbance of either form at 565 nm yet the Z form is recovered after 2 seconds of irradiation. Can the authors comment on this? Is it simply that, although there is very little absorption of either form at 565 nm, E does absorb more strongly than Z? If so, this is hard to see from the data.

(3) Following on from comment (2), it would be easier to assess these if molar absorption coefficients for Z and E forms were given at the wavelengths of irradiation. It would then be easier to compare the effect of band separation and quantum yields on PSS.

(4) HPI-3-OH does retain some switching properties, and they may well still make the switch suitable for biological applications. However, they are quite different from the phosphinate ester. Are the authors able to suggest why the properties might be so different? Given that ester to acid has generated a significant change in properties, might it be expected that other ester groups might have interesting effects as well?

Very minor comments:

Line 96 "wasevidenced". Add a space.

Line 109 Perhaps the authors mean HPis 8 and 9 rather than 7 and 8?

Version 1:

Reviewer comments:

Reviewer #1

(Remarks to the Author)

I would like to thank the authors for additional efforts and detailed responses to all concerns raised by me and other reviewers. The revised version of the manuscript has been significantly improved and enriched with important information on

photoswitching of HPis in water. In my opinion, the manuscript can now be accepted for publication in Nat. Commun.

Reviewer #2

(Remarks to the Author)

The authors carefully addressed all comments of three reviewers, and the manuscript has been satisfactorily improved. I think this manuscript has met the requirements and recommend this work for publication in Nature Communications without further revisions.

Reviewer #3

(Remarks to the Author)

I believe the authors have satisfactorily addressed the comments in my original review, and support publication of the manuscript in its revised form. I note the authors have made considerable changes to the manuscript in response to the other reviewers' comments, and the paper is an important contribution to the photoswitch literature.

Answers to Reviewers

We thank all Reviewers for their extremely helpful and insightful questions and requests, which allowed us to significantly improve our work and its presentation. With the Reviewers help we were able to fully unlock the high potential of the HPis, especially as water-soluble photoswitches, and show their elevated performance. Below we answer to all Reviewer comments in detail.

Reviewer 1:

The manuscript by Dube and co-workers describes synthesis and photoswitching of hemiphosphoindigo derivatives. Importantly, this class of compounds has been described previously but the photoswitching has not been studied yet. In general, the manuscript could be interesting for the readership of Nat. Commun. and fitting to the scope of the journal. However, in my opinion, publication of this work is premature and further experiments are needed to make the manuscript ready for publication. Below are the points that I would like the authors to consider:

Answer:

We would like to thank Reviewer 1 for the very supportive evaluation of our results and for highlighting the fit of these results for Nat. Commun. Thanks to the many insightful and detailed points, we were able to significantly improve this work and actually unlock the full potential of HPis, especially with respect to in water photoswitching. This was very great advice and we are happy that we were asked to scrutinize the HPI performance further, which enabled us to showcase their excellent properties comprehensively.

Reviewer 1:

1) The statement “water soluble photoswitches” in the title and in the text is misleading. Only three out of 18 compounds have been checked for the photoswitching in water and two out of these three ones showed either bad (42% conversion in pss) or very bad (18% conversion in pss) performance in water. Additionally, the choice of compounds for the studies in water looks rather random (one twisted HPI, two heterocyclic ones, no planar ones...). To explore the photoswitching of HPI in water, the solubility and photochromism should be checked for all 18 compounds and honestly compared with the literature data on other classes of photoswitchable molecules in water (for collection of examples, see <https://doi.org/10.1039/D0CS00547A>).

Answer:

Thank you very much for raising this important point. We agree with the Reviewer that the photoswitching in water was somewhat scattered and not complete. We have now comprehensively investigated the photoswitching of all HPIs 1-18 in water and added a full chapter to the revised SI (1.9 Photoisomerization reactions in water). Solubility was first qualitatively investigated via UV/Vis spectroscopy. For selected promising derivatives, water solubility was then also quantified and added to Table S1 in the SI (1.3 Water solubility). We further investigated a dedicated selection of HPI derivatives in their hydrolyzed form, i.e. HPIs 1-OH, 2-OH, 11-OH and 17-OH. This selection was based on the favorable photoswitching properties and photochromism of the corresponding phosphinate derivatives in toluene solution. We are very happy that Reviewer 1 insisted on this point as we are now able to demonstrate significantly improved performance of HPI photoswitching in water. We now achieved very good photoswitching reaching up to 78% E isomer enrichment and 100% Z isomer enrichment in the corresponding pss (see newly added Table 2 in the manuscript). As their water solubility is also quite good, with up to 0.85 mM solutions reachable in pure water, we can thus state that HPI photoswitching in water is very well competitive with the state of the art in the literature, as summarized nicely e.g. in the reference given by Reviewer 1, which we also now cite.

We thus revised this manuscript section completely and incorporated our new findings accordingly. The results for 3-OH, 15 and 17 in water have been moved to the corresponding new chapters accordingly.

Reviewer 1:

2) Abstract: "... high isomer enrichment..." – this statement cannot be applied to the most of the studied compounds (<80% of E-isomer in PSS).

Answer:

Thank you for this comment, indeed not all photoswitches presented in this study display favorable photophysical properties. The sentence was therefore rephrased to "Phosphinate is embedded in the core indigoid chromophore and the resulting optimized photoswitches display high thermal stabilities, excellent fatigue resistance and high isomer enrichment in the photostationary states."

Reviewer 1:

3) Lines: 44-46 and 142-143: the information on the recent studies of the photoswitching of aurones belonging to the indigoid family should be added and the corresponding publications should be cited.

Answer:

Thank you for this point, the information was added and the corresponding publications were cited explicitly mentioning aurones now in the introduction.

Reviewer 1:

4) Lines 63-65: could the authors explain why no acceptor groups were introduced in the twisted HPI?

Answer:

Planar HPI 12-14, bearing acceptor groups, display decreased resistance to photofatigue, which can be seen in the corresponding section in the SI. This behavior is in line with previous studies on HTIs in our group. For this reason, no acceptor groups were introduced in the twisted HPIs. We added the following clarification to the text: "No acceptor substituents were introduced into twisted derivatives as acceptor groups generally proved to be detrimental for the performance of hemiindigoid photoswitches.⁵⁰"

Reviewer 1:

5) It is not clear, whether all presented HPI are novel compounds or some of them have been reported previously. Please add this information to the text.

Answer:

Thank you for raising this important point. All derivatives except for unsubstituted HPI 11 are novel compounds. We have added this information to the revised manuscript by writing: "To the best of our knowledge all here presented HPIs with the exception of HPI 11⁹⁸ are described for the first time in this work."

Reviewer 1:

6) Since the photoswitching of HPI is reported for the first time, it is necessary to isolate and fully characterize the photoinduced *Z*-isomers. Considering the long lifetimes, this should be possible for most of the compounds. If this is done, the molar absorption coefficients and absorption spectra of the photoinduced isomers can be determined directly (cf. the calculations described in Section 1.4 of SI).

Answer:

Thank you for this comment. We believe that the reviewer refers to the metastable *E* isomers, since the *Z* isomers have been fully characterized throughout this work. We do however not fully agree with the Reviewer at this point. Solid structural proof for the stable *Z* isomers have been given through full NMR spectroscopic analysis, including the assignment of all signals and in form of crystal structures. All photoisomerizations were followed by ¹H NMR spectroscopy and for each signal of the *Z* isomer a corresponding *E* isomer signal can be detected. Together with the vicinal coupling constant analysis in Table S7 and the direct NOE NMR analyses conducted, our spectroscopy data provide ample and adequate proof that the structural change induced upon light irradiation is indeed double-bond isomerization to the metastable *E* isomers. The determination of molar absorption coefficients and absorption spectra from mixtures is a well-established method in the field and has proven reliable time and again in numerous studies at all journal impact levels and by all eminent groups in the field (see for example: <https://doi.org/10.1039/D4SC00983E>, <https://doi.org/10.1021/jacs.9b06104>, <https://doi.org/10.1021/jacs.7b08726>). Adding physical isomer separation schemes for all 18 derivatives investigated here would seriously delay publication of our work and would not add any new information. To prove this point, we have now isolated the *E* isomer of HPI **1** and fully characterized it again independently (see chapter 1.2 HPI synthesis). Figure S26 (chapter 1.5.2) in the SI shows a comparison of the molar absorption coefficients determined from the mixture and from the pure isolated isomer. Both spectra are in very good agreement and show minor deviations in intensities that are well within the range observed by repeated independent measurements. This finding clearly supports our claim that isolation is not necessary to determine the molar absorption coefficients or other analytical data for the metastable *E* isomers adequately.

Reviewer 1:

7) Line 175: "... very fast... photoswitching..." - please add the corresponding reaction times to Table 1.

Answer:

Thank you for this point. While it is important to mention the irradiation times for the photoreactions, the times are only comparable to a limited extent, since they depend on many factors such

as the LED used, the distance between LED and sample, and the concentration of the sample. For better comparability, also with photoswitches from other groups, quantum yields were chosen as a measure for efficiency since they are independent from the factors mentioned above. For maximum reproducibility, the LEDs used and all irradiation times can be found in the SI. However, we acknowledge the Reviewer calling out that when making a statement about “very fast photoswitching” we should give a rough estimate. We thus changed the following sentence in the revised manuscript to address the Reviewers comment directly: “The quantum yields are quite high and thus allow for very fast and efficient photoswitching within seconds to minutes under typical low- and medium power LED irradiation at UV/vis concentrations (see also chapter 1.8 in the Supporting Information).”

Reviewer 1:

8) Lines 256-259: “... fully suited for many biological applications...” Please provide examples of the relevant biological applications along with references.

Answer:

Thank you very much for this point. We have now added references where self-deactivation of a photopharmacophore is described in order to fully control a biological effect. We added: “(for a discussion of thermally labile photoswitches for biological applications see for example Ref. ¹⁰⁹⁻¹¹¹ and literature cited therein)”

Reviewer 1:

9) Taking into account the above critical points, the conclusions are not fully supported by the results and should be revised.

Answer:

Thank you very much we have revised also the Conclusion section to accommodate our newly delivered results as well. With these results we do stand by our claims, that HPs are very versatile and valuable new photoswitches with different applicabilities in different areas of research.

Reviewer 2:

In this paper, the authors have explored a class of phosphorus-based indigoid photoswitches with high thermal stabilities, excellent fatigue resistance, and very good switching performance,

which could be used in the field of data storage, sensing. Further, the molecular framework allows facile late-stage conversion to water-soluble photoswitches with a biologically group, while maintaining the switching properties. This has high potential for the development of light-activated drugs and other biological applications in the future. In my opinion, this work is sufficiently novel and suitable to be accepted by Nature Communication, but there are some problems in the manuscript that need the authors make changes after considering the following suggestions.

Answer:

We would like to thank Reviewer 2 for this very positive and supportive assessment of the novelty and importance of our work and for supporting its publication in Nat. Commun.

Reviewer 2:

Suggestions noted below:

1. How to prove that Z-3 adopts twisted state in solution?

Answer:

Thank you for this insightful comment. To best prove the twisted state of Z-3 one can observe the chemical shifts of the ethyl group of the phosphinate in the ^1H NMR spectra. For planar HPis the CH₂ group signal is consistently found at chemical shifts in the range of 4.05 to 3.90 ppm and the CH₃ group is found consistently at 1.22-1.18 ppm. For twisted HPis the corresponding shifts are significantly different and upfield shifted as it is expected from the localization of the ethyl group above the ring current of the twisted aromatics. Here we observe for twisted derivatives consistently CH₂ group signal shifts in the range of 3.86-3.43 ppm and for the CH₃ group in the range of 1.15-0.83 ppm. Since Z-3 shows very clearly the same behavior as for the twisted HPis (CH₂ and CH₃ signal shifts of 3.77 /3.62 and 1.14, respectively) we are very confident that it is in fact twisted in solution. A similar analysis of aromatic ring-current induced signal shifting in HTIs to evidence twisting in solution can be found in [doi: 10.1021/jacs.6b05981](https://doi.org/10.1021/jacs.6b05981) and [doi: 10.1039/C9QO00202B](https://doi.org/10.1039/C9QO00202B).

Support for this structural assignment comes from our quantum chemical calculations including a PCM solvent model, which show again twisted conformations to be most stable for Z-3.

We added this analysis to the revised Supporting Information (Table S8 in chapter 1.8) and also added a new corresponding section in the revised manuscript.

Reviewer 2:

2. What is the relationship between Figures S3, S4 and the order of thermal isomerization as described in line 104?

Answer:

Thank you for this question, this was in fact a mistake and we have corrected the order of the Figures accordingly. For a unimolecular first order reaction, the logarithmic plot (c) should give a linear relationship (according to Equation S3) as can be seen for the other derivatives. Since this is not the case for HPI **1** and **2**, this indicates that the thermal back isomerization is not a first order reaction but a more complex process and cannot be described by such kinetic analysis. As we do not know the order of this reaction and different orders could be fitted to it, we refrained from speculation and just report the half-lives as somewhat quantitative measures.

Reviewer 2:

3. What are the possible reasons for the high water solubility of Z-15 and Z-17?

Answer:

Thank you very much for this question. The derivatives with imidazole substituent have strong tendencies to form hydrogen bonds with water and are therefore expected to increase solubility. Moreover, their polar/apolar group ratio is higher due to their comparatively smaller molecular size and the larger number of heteroatoms overall.

Reviewer 2:

4. Please provide the light irradiation time required for photoisomerization of the three photoswitches shown in Figure 2.

Answer:

Thank you for this comment. Irradiation times were added to Figure 2.

Reviewer 2:

5. In the captions of Figures S3 and S4, it states, "the value is given in Table S1," but Table S1 contains data on water solubility. Please check them.

Answer:

Thank you very much for pointing out this mistake, which is now corrected in the revised SI.

Reviewer 2:

6. The geometry of HPIs 15-18 should also be provided in Supporting Information for the reader's reference for comparison.

Answer:

Thank you for this comment. No crystal structure was obtained for this derivative but the optimized geometries of HPI 15-18 were added in chapter 4 of the Supporting Information. The calculations indicate mostly planar geometry with slight orientation of the N-H or (C-H in the case of HPI 18) to the oxygen of the phosphinate.

Reviewer 2:

7. On line 73, why crystals sample suitable for X-ray structural analysis can't be obtained from HPI derivatives Z-6, 7, 11, 13-18? Whether it is the poor chemical stability of these products under X-rays or some other problem? Please the authors explain this in the main text.

Answer:

Thank you for this point. Unfortunately, we were not successful to obtain crystals suitable for X-ray structural analysis. Compounds are stable but do not crystallize well enough apparently. In most case the problem was that the substance only yielded amorphous solids. Moreover, HPI 16 is an oil.

Reviewer 2:

8. On line 108, HPI 7 is not planar geometry. In addition, the comparison between HPI 7,8 and HPI 10-14 can't explain the trend authors referred before. Please the authors correct it.

Answer:

Thank you for pointing out this numbering error. The authors meant HPI 8 and 9. This has been corrected in the manuscript. The trend is now clearly observable.

Reviewer 2:

9. On line 142-145, the data here are insufficient, and the authors should convert some of the examples from HTIs, HIs, and HPis into figure to more visually show the significant blueshift of HPis.

Answer:

Thank you for this comment. We added two new Figures to the revised SI where the molar extinction coefficients of methoxy- derivatives of HTI, Sulfoxide-HTI, Sulfone-HTI (in dichloromethane solution) and Julolidine-derivatives of HI, HTI, Sulfoxide-HTI, and Sulfone-HTI (in toluene solution) are compared to the respective HPis.

Reviewer 2:

10. On line 245-267, the necessary figures to show the data are missing here. The data for HPI 15-OH and HPI 17-OH should be supplemented and the data for HPI 3,15,17, HPI 3-OH,15-OH,17-OH should be compared.

Answer:

Thank you very much for this point. We have now comprehensively addressed the water solubility of both, phosphinate HPis and selected hydrolyzed HPis and rewritten the chapter completely. We also added a large body of new experiments and data to the SI for a much more complete investigation. As in our answer to Reviewer 1 regarding this point:

We have now comprehensively investigated the photoswitching of all HPis 1-18 in water and added a full chapter to the revised SI (1.9 Photoisomerization reactions in water). Solubility was first qualitatively investigated via UV/Vis spectroscopy. For selected promising derivatives, water solubility was quantified and added to Table S1 in the SI (1.3 Water solubility). We further investigated a dedicated selection of HPI derivatives in their hydrolyzed form, i.e. HPis 1-OH, 2-OH, 11-OH and 17-OH. This selection was based on the favorable photoswitching properties and photochromism of the corresponding phosphinate derivatives in toluene solution. We are very happy that Reviewer 1 insisted on this point as we are now able to demonstrate significantly improve performance of HPI photoswitching in water. We now achieved very good photoswitching reaching up to 78% E isomer enrichment and 100% Z isomer enrichment in the corresponding

pss (see newly added Table 2 in the manuscript). As their water solubility is also quite good, with up to 0.85 mM solutions reachable in pure water, we can thus state that HPI photoswitching in water is very well competitive with the state of the art in the literature, as summarized nicely e.g. in the reference given by Reviewer 1, which we also now cite.

We thus revised this manuscript section completely and incorporated our new findings accordingly. The results for 3-OH, 15 and 17 in water have been moved to the corresponding new chapters accordingly.

Reviewer 2:

11. If possible, the authors provide a potential application case, which could be used in the field of data storage or sensing.

Answer:

Thank you for this point, we have now added appropriate references about photoswitch applications in optical memories and sensing to answer to this point.

Reviewer 3:

The manuscript describes an extensive series of new photoswitches comprising hemiphospho-indigo cores with varying substituents on the stilbene moiety. This further diversifies the extensive work this group has done on indigoid-type photoswitches. The short synthetic route enables a variety of derivatives to be made relatively easily. The synthesis and characterisation of the switches is presented clearly and thoroughly, and detailed photophysical and photochemical characterisation has been carried out to develop structure/property relationships for the various observations. Efforts have been made to support these conclusions with computational work, although the authors note that for some of the switches, conformational variation makes spectral prediction by TDDFT challenging.

The photoswitches are interesting new additions to the literature, and the added aspect of the chiroptical properties makes them an exciting area for further development.

I think the manuscript is suitable for publication, subject to a few minor clarifications.

Answer:

Thank you very much for your very positive assessment of our work and for supporting its publication in Nat. Commun., we greatly appreciate it!

Reviewer 3:

Minor comments:

(1) Table 1 might benefit from extra columns to separate some of the data. The current layout is quite difficult to follow.

Answer:

Thank you for this point, we have tried to add a lot of information into this table and we are not quite sure how to improve readability. Please also note that Nat. Commun. inherits a distinct table layout, where the different column entries are shaded differently to improve readability. We thus believe that the layout of the Table will be much improved in the edited version of our manuscript.

Reviewer 3:

(2) Figure 2 and line 165 onwards – the authors say that planar HPis 8-14 have very little band separation hence poor Z to E conversion on irradiation. But for some compounds, this still seems to result in almost quantitative recovery of the Z form despite similar quantum yields for forward and reverse processes. e.g. HPI-8, the PSS achieved under 565 nm irradiation is almost quantitative in Z, despite the quantum yields for Z to E and E to Z being equal. There appears to be no appreciable absorbance of either form at 565 nm yet the Z form is recovered after 2 seconds of irradiation. Can the authors comment on this? Is it simply that, although there is very little absorption of either form at 565 nm, E does absorb more strongly than Z? If so, this is hard to see from the data.

Answer:

Thank you for this insightful comment. The *E* isomers all display red-shifted tailing in the absorption spectrum and have a region, where the *Z* isomer does not absorb at all but the *E* isomer still absorbs a little. This difference is enough for the photoreaction to quantitatively occur. For example in mentioned HPI 8 this is the case at around 540 nm. Since LEDs and not lasers are used for irradiation the emission spectrum of the LED is not a sharp line but a Gaussian-shaped band centered around the maximum indicated for the LED. This means that the edge of the 565 nm LED will be at around 540 nm and exactly irradiate the indicated region without hitting the *Z* isomers. For illustration we have now added an enlarged zoomed-in region of the absorption spectra of both isomers of HPI 8 to revised Figure S34 in order to make this absorption difference in the bathochromic region of the spectra clearly visible.

Reviewer 3:

(3) Following on from comment (2), it would be easier to assess these if molar absorption coefficients for Z and E forms were given at the wavelengths of irradiation. It would then be easier to compare the effect of band separation and quantum yields on PSS.

Answer:

Thank you very much for this point. Following the explanation from point (2), the absorption at the wavelength (maximum) of irradiation indicated on the LED does not necessarily give better information since the full emission spectrum needs to be taken into account and then should be overlaid with the full molar absorption profile of both isomers for every nominal LED used for irradiation. We have reported the LED emission spectra already in the past and published these data ([doi: 10.1021/jacs.7b07531](https://doi.org/10.1021/jacs.7b07531), [doi:10.1002/anie.202409214](https://doi.org/10.1002/anie.202409214)) and we believe that showing the absorption spectra and clearly stating which LEDs were used provides all information needed to reproduce our results and explain the observed behavior. In order to make this easier for the readers we added the following comment and references to the revised SI: "The isomer composition in the pss is generally a function of quantum yields for individual transformations of both isomers and their spectral overlaps at a given wavelength of irradiation. Since LEDs were used for irradiation their corresponding emission profiles have also to be taken into account instead of just their nominal maximum wavelengths, which can lead to deviations in the observed pss compositions from the expected ones at a single wavelength. Information about the used LEDs emissions can be found e.g. here Ref. [8,13]"

Reviewer 3:

(4) HPI-3-OH does retain some switching properties, and they may well still make the switch suitable for biological applications. However, they are quite different from the phosphinate ester. Are the authors able to suggest why the properties might be so different? Given that ester to acid has generated a significant change in properties, might it be expected that other ester groups might have interesting effects as well?

Answer:

Thank you for this comment. We believe that there are two major effects coming into play:

First, the electronic effect. The phosphinate is a slightly weaker electron acceptor as opposed to the phosphinic acid due to the +I effect of the alkyl group. For Hemiindigoid photoswitches in general it is beneficial to have a push-pull indigoid system consisting of an electron donor and an

electron acceptor at the indigoid fragment to give red-shifted absorptions and substantial photochromism. Usually, the carbonyl functions as acceptor and the heteroatom as the donor. In the case of the HPIs, the phosphinate group serves as a second acceptor, resulting in less pronounced photochromism and red-shift. The phosphinic acid, being a stronger acceptor, follows this trend further and worsens the switching properties. The best result would be obtained with a trivalent phosphorus (which is prone to oxidation). Introduction of electron donating alkyl groups at the ester could indeed improve photoswitching properties by making the acceptor weaker and we gladly take up this suggestion of Reviewer 2 in a follow-up study.

Second, the solvents are very different (water and toluene). Usually, large shifts in absorption can be seen upon changing the solvent (solvatochromism). As phosphinic acids are also quite acidic we cannot exclude that acid-base properties and significant hydrogen bonding effects would further change the photochemistry properties. For example, it might be possible that proton transfer reactions take part in the excited state chemistry. As these possible causes are nevertheless still speculative we refrained from adding this discussion to the revised manuscript and SI.

The diminished thermal stability of HPI 3-OH is most probably also a result of the acidity. This phenomenon is seen time and again in hemiindigoids, where the presence of protons leads to markedly reduced energy barriers for thermal double bond isomerizations. In indigo itself just the water content alone and its capacity for hydrogen bonding leads to dramatic changes in the thermal stability of the metastable cis-isomer as shown here: [doi: 10.1002/cptc.201700228](https://doi.org/10.1002/cptc.201700228)

Reviewer 3:

Very minor comments:

Line 96 "wasevidenced". Add a space.

Line 109 Perhaps the authors mean HPIs 8 and 9 rather than 7 and 8?

Answer:

Thank you very much for pointing out these errors. They were corrected in the revised manuscript.